# Fruit and Non-Starchy Vegetable Acquisition and Supply in Solomon Islands: Identifying Opportunities for Improved Food System Outcomes

Penny Farrell [1,*], Michael K. Sharp [2,3], Erica Reeve [1,4], Tom D. Brewer [3], Anna K. Farmery [3], Jillian Tutuo [5], Jessica R. Bogard [6], Samson Kanamoli [7] and Anne Marie Thow [1]

1   Menzies Centre for Health Policy and Economics, The University of Sydney, Sydney, NSW 2050, Australia
2   Pacific Community, Noumea 98848, New Caledonia
3   Australian National Centre for Ocean Resources and Security, University of Wollongong, Wollongong, NSW 2522, Australia
4   Institute for Health Transformation, Faculty of Health, Deakin University, Geelong, VIC 3216, Australia
5   WorldFish, Honiara P.O. Box 438, Solomon Islands
6   Agriculture and Food, Commonwealth Scientific and Industrial Research Organisation (CSIRO), St. Lucia, QLD 4067, Australia
7   National Statistics Office, Ministry of Finance & Treasury, Honiara P.O. Box G6, Solomon Islands
*   Correspondence: penny.farrell@sydney.edu.au

**Abstract:** The Pacific Guidelines for Healthy Living recommend consuming a minimum of five servings of fruit and/or non-starchy vegetables each day, however, diets in Solomon Islands stray from the regional and global trend of healthy diets high in fresh fruit and vegetables. Our study drew on multiple sources of data and a food systems framework to show a country-wide picture of the role and benefits offered by fruit and non-starchy vegetables in Solomon Islands. First, we analysed data on fruit and non-starchy vegetable consumption and matched this to the data on supply. Second, we used a policy documentary analysis to highlight opportunities for the roles of fruit and non-starchy vegetables in the Solomon Island food system to advance progress in multiple Sustainable Development Goals. Key findings related to supply were the findings that domestic production of fruit and non-starchy vegetables is insufficient to meet per capita requirements, which coupled with our finding that per capita national level supply through imports is inconsequential, thus highlighting important undersupply issues for the nation. The food environment analysis indicated multiple further challenges hampering fruit and non-starchy vegetable consumption. Integrated with our analysis of policy, these revealed several opportunities, including improving affordability of this healthy commodity, enhancing livelihood equitability of supply chains, and strengthening environmentally sustainable agricultural practices that support increased production.

**Keywords:** fruit; vegetables; Solomon Islands; Pacific; food system; policy

## 1. Introduction

Increasing fruit and vegetable consumption as part of a balanced diet provides opportunities for synergistic gains for public health, environmental sustainability, and livelihoods [1,2]. The United Nations General Assembly declared 2021 the International Year of Fruits and Vegetables in recognition of the vital role that fruit and vegetables play in food security and health. Their role across the food system has been linked to at multiple United Nations Sustainable Development Goals (SDGs) [3]. Shifting global diets away from red meat and unhealthy processed foods and towards diets high in healthy plant-based foods is a recognised opportunity to help reduce the negative environmental impacts associated with food systems, including climate change, loss of biodiversity, use of freshwater, land-system change, and interference with the global nitrogen and phosphorus cycles [1]. In terms of public health gains, increasing consumption of fruit and non-starchy vegetables

can reduce the risk of developing non-communicable diseases (NCDs) and micronutrient (including Vitamin A, iron, folate, and calcium) deficiencies [1]. There is global evidence that optimal fruit and vegetable consumption reduces the risk of heart disease [4,5], and fruit and non-starchy vegetable (FNSV) consumption can significantly help reduce the risk of developing diabetes [6–10], unhealthy weight increase [10], mental illness [11,12], and cancer [13].

In Solomon Islands, a country in the South-West Pacific with an increasing population of currently just over 700,000 people [14], the majority (75–80%) of people live in rural areas where subsistence diets consist of locally grown and sourced plant-based foods and fresh fish [15–17]. FNSV currently provide the population with an intake of around two-thirds of vitamin A, and three-quarters of beta-carotene, and one-fifth of vitamins B and C and calcium, with main sources of these micronutrients being cabbage and papaya [16,18]. Diets across Solomon Islands have been following the regional transition away from diets high in locally produced and harvested fresh vegetables and fruit, seafood, nuts, and starchy root crops for the past half-century [19,20]. Instead, diets are increasingly characterised by consumption of calorie-dense, often imported, processed foods that are high in fat, refined sugar, and salt, and low in nutrients; this is particularly the case in urban areas [15,16,21,22]. These foods are majorly contributing to the high and increasing burden of obesity and NCDs such as cardiovascular disease and diabetes [23–27]. The nutrition transition is manifesting as a triple burden of malnutrition: rates of overweight (SDG 2.2.2), stunting (SDG 2.2.1), and micronutrient deficiencies, including prevalence of iron deficiency anaemia amongst women of reproductive age (SDG 2.2.3), are concerning [15]. Solomon Islands has the highest reported rate of undernourishment (SDG 2.1.1) in the Pacific region [28], and one-third of children experience stunted growth [21,29]. Diabetes is now the leading cause of death and disability in Solomon Islands, and diabetes care accounts for at least 20% of the government's annual health care expenditure [30]. This is placing a significant strain on an already over-burdened health system [21,31]. This burden of disease in the population is placing direct and indirect pressure on the economy, and leading to significant personal and societal loss from early morbidity and deaths [17].

The World Health Organization (WHO) and the Pacific Guidelines for Healthy Living endorse the consumption of at least five servings of fruit and/or vegetables per day, which equates to 400 g total—excluding starchy vegetables such as cassava and sweet potato [32,33]. However, as with most countries around the world [34,35], consumption of FNSV in Solomon Islands does not align with the dietary guidelines. The most recent STEPwise approach to surveillance (STEPS) survey shows that 93% of the population ate less than the recommended serve of FNSV per day [36], and an analysis of the latest Solomon Islands Household Income and Expenditure Survey (HIES) shows that the per person average FNSV consumption is less than half of the recommended intake [16].

As in all countries, the reasons for insufficient FNSV consumption in Solomon Islands are complex [34]. Solomon Islands Government policies have consistently applied health promotion approaches to encourage consumption of local food, or 'lokol kaikai' [37], however sustainably enabling adequate, available and affordable supply of FNSV to populations requires coordination of action across sectors beyond the health sector, in particular sectors focussed on agriculture, industry, economics, and trade [38]. Identifying sustainable opportunities for improvements that balance population health, livelihood, and environmental outcomes requires analysis of food systems-oriented data and policy on food production, supply chains, food environments, and food acquisition and consumption [2,38]. Efforts to increase FNSV production, for example, will need to contend with challenges such as increased invasive pests and diseases, safe use of agrochemicals, and declining soil fertility and biodiversity, as a result of intensive land use, deforestation, and mining [39]. The Solomon Island food system is also prone to shocks related to natural disasters, which are predicted to become more frequent and severe with climate change, civil unrest, and most recently the COVID-19 pandemic, and these have ongoing impacts which are important to understand through a whole-food system lens [40].

While the health benefits of increasing FNSV consumption are clear, there are complexities in ensuring that changes in food systems are sustainable. Food systems that are not equitable contribute to poverty, malnutrition, and food insecurity, especially for vulnerable populations including children and women [23]. A food system framework provides a structure to assess and measure FNSV in a holistic sense, and therefore provides tangible opportunities for improvement in environmental and population health outcomes, poverty reduction, and food security by enabling leverage points to be identified across the system. As the concept of food systems has increased in prominence over the past decade, the number of food system conceptualisations has risen quickly. In line with this, researchers are seeking to identify how best to measure food systems and map them to food system outcomes related to human health, environmental sustainability, and livelihoods, as well as cross-cutting themes such as governance and resilience [2]. In this study, we draw on a 2021 framework, which conceptualises food systems as comprised of food supply chains (including food production and inputs, food storage, loss, distribution and transport, retail, markets, and waste), and food environments (including food availability and affordability)—which in turn influence individual behaviours including food acquisition [2]. Food system drivers include biophysical and environmental drivers, politics and leadership, sociocultural dynamics, and globalization and trade. Outcomes, or outputs of food systems, include those related to nutrition and health, the natural environment, economies, and social equity and inclusion [2].

In this paper, we describe key aspects of the food system with respect to FNSV in Solomon Islands, including production, trade, and food environments. Production of FNSV within Pacific Island Countries and Territories is currently too low to supply the amount of FNSV required to be consumed for good health [41], and there is a need for analysis at smaller scales to understand sub-national patterns in availability [41]. The aim of this study is to identify possible pathways for increasing FNSV consumption to improve human health outcomes while considering environmental sustainability and identifying opportunities to meet corresponding SDGs. To address this broad aim we use a contemporary food system framework [2] to analyse data on: (a) consumption through the food environment, to understand how dietary behaviour is influenced by food environment sub-components including location and income status, (b) FNSV supply including local markets, production and trade, and (c) government policy on FNSV acquisition and supply in Solomon Islands.

## 2. Methods

### 2.1. Study Design

This study applies a consumption-oriented approach [42] to analyse key elements of FNSV supply and acquisition in Solomon Islands from the point of household acquisition back through to production or harvest, including imports. It uses a contemporary food systems framework [2] to define problems in a data-informed way across the food system, to map them to policy, and identify leverage points to increase sustainable consumption of this healthy food group.

We drew on multiple sources of data to show a whole-country picture of FNSV acquisition and supply in Solomon Islands. First, we analysed data on food use (consumption from the most recent HIES) and insights on distribution and unaccounted for use, such as loss and waste data from a survey conducted by the author team on FNSV market supply in 2020 and 2021). We then matched this against data on supply (production from the FAOSTAT database and net imports from the Pacific Food Trade Database) (Table 1). Second, we used a policy documentary analysis approach to examine food system policy commitments relevant to FNSV supply and acquisition to identify opportunities and focus areas for FNSV related policy and programs across the food system. The data from the different sources (databases, surveys and policies) were analysed separately, then the key findings and implications were interpreted by the authors, whose expertise spans across nutrition, food systems, policy, statistics, food environments, and sustainable agriculture.

**Table 1.** Overview of food system data sources and analysis.

| Components of Food System ^ | Data Source | Data | Data Analysis | Key Relevant SDG's |
|---|---|---|---|---|
| Food supply chains | | | | |
| Production | FAOSTAT [43] | FNSV production | Quantitative descriptive: timeline | 1, 2, 3, 4, 5, 8, 11, 12, 15 |
| Trade | Pacific Food Trade Database [44] | Net imports of FNSV | Quantitative descriptive: timeline | 2, 8 |
| Retail, markets & waste | Market vendor survey and HIES [45] | Interviews about FNSV supply; Consumption expenditure by food environment type | Vendor survey: Qualitative thematic HIES: quantitative descriptive using Bogard et al. 2021 typology | 2, 5, 8, 12–15 |
| Food environments and Individual factors | | | | |
| | HIES [45] | Consumption expenditure and apparent nutrient consumption | Quantitative multivariate | 1, 2, 3, 5, 10 |

^ Food system components from food system framework in Figure 1 in Fanzo et al. 2021.

### 2.2. Individual and Food Environment Factors Which Affect Acquisition

2.2.1. Individual Economic, Situational, and Affordability Analysis Using Household Income and Expenditure Survey Data

Data on food consumption expenditure (a proxy for food consumption) from the Solomon Islands 2012–13 Household Income and Expenditure Survey (HIES) [45] were analysed to describe FNSV household expenditure. Multivariate analysis was performed to characterise populations with adequate expenditure to acquire a sufficient quantity of FNSV to meet consumption thresholds set by the WHO (400 g per adult per day).

The HIES collects a comprehensive set of demographic, gender, health, and economic information on individuals, and a broad spectrum of household level data. The HIES adopted a stratified sampling methodology using the 2009 census frame, including rural and urban areas of each province. The data were weighted to be nationally representative. A sample size of 4478 households was included. In the HIES, food expenditure (a proxy for acquisition) information is collected from individual households captured through a 2-week diary that records food expenditure quantity and value by differing source—cash purchases, home production, and in-kind (gifts). We included all at-home food expenditure, which was classified in the HIES dataset as: (i) cash purchases, (ii) home production, and (iii) in-kind (gifts) received.

The dependent variable was whether or not household expenditure was sufficient to acquire 400 g FNSV per adult per day, following the method of Jones and Charlton (2015) [24]. This variable was calculated by dichotomizing households as having sufficient or insufficient FNSV consumption based on their daily required expenditure to obtain 400 g per adult equivalent per day. Consumption data on FNSV, consisting of 55,207 individual acquisition transactions covering 22 fruits and 29 non-starchy vegetables, were converted into edible portions using conversion factors sourced from the Pacific Nutrient Database [46]. Value price per edible gram was derived (and cleaned by identifying outliers as being three-times the interquartile range beyond the mean of the logged consumption distribution, by FNSV item and region, and treated as the median by the same groups). The lowest-cost FNSV basket was derived by region, and each consisted of two 80 g portions of fruit and non-starchy vegetables and another 80 g portion of the next lowest cost fruit or non-starchy vegetable item to make up the recommended five servings per adult person per day.

Income, proxied by expenditure, was calculated as above or below median per adult equivalent (top 50% vs. bottom 50% wealth). The variable for household size was dichotomized using the mean which was just under 6. The top and bottom 50% income variable was created by converting household income to Adult Male Equivalent income

and using the weighted median income to dichotomise households to top and bottom groups. Descriptive statistics for these variables are included in Table A1, Appendix A.

A multi-logistic analysis was conducted, following these steps: Step 1. bivariate analysis models with each independent variable (as listed in column of Table 2 below) and the outcome (dependent) variable (Table A2); Step 2. multivariate analysis including those variables that returned $p$-values $\leq 0.1$ in the bivariate analysis (Table A3); Step 3. a final, "best fit" multivariate analysis model including the variables that had returned $p$-values $\leq 0.1$ in Step 2. Complex sampling design was accounted for in the analysis, and outputs are representative of the total population. The analysis was performed using Stata/SE 15.1 (Stata Corporation, College Station, TX, USA). This secondary analysis on anonymised data from the 2012-13 Solomon Islands HIES dataset was officially exempted from review by the University of Sydney Human Research Ethics Committee.

**Table 2.** Individual and food environment factors included in multivariate analysis.

| Individual and Food Environment Factors | Input Variable | Reference Category |
|---|---|---|
| Age of household head | Aged 40 and above | 15–39 years |
| Sex of household head | Male | Female |
| Highest education attainment of household head | Senior secondary and tertiary | Preschool to junior secondary |
| Marital status of household head | Married | Not married |
| Household has a wage income | Has wage income | Does not have wage income |
| Household participates in agriculture | Participates in agriculture | Does not participate in agriculture |
| Household participates in fishing | Participates in fishing | Does not participate in fishing |
| Household size | 7 or more | 6 or less |
| Income | Top 50% | Bottom 50% |
| Region | Urban | Rural |
| Has a vegetable garden | Household has a vegetable garden | Household does not have a vegetable garden |

2.2.2. Vendor Properties and Food Availability Analysis Using Household Income and Expenditure Survey Data

Household consumption data from the 2012–13 Household Income and Expenditure Survey dataset were analysed according to the Pacific food environment typology to derive descriptive statistics showing the proportion of FNSV acquired from each food environment type (retail, cultivated, wild, kin and community) [47]. Ethical approval was received from the CSIRO Social and Interdisciplinary Science Human Research Ethics Committee (035/21).

*2.3. Food Supply Chains*

2.3.1. Local Markets and Production Analysis Using Vendor Survey of Urban Food Markets

Vendor surveys were performed twice: first in July to August 2020, then in July 2021, in Auki, Gizo, and Honiara. The study recruited vendors within open air markets and impermanent vendors selling from the roadside intersection or temporary shelters. The survey was conducted on market days, which included some weekend days. Data were collected using a survey tool that collected information on food supply and sourcing of food commodities. Participants' responses were recorded on tablets programmed with the data collection template using the program KoBoToolbox [48]. Thematic analysis

was used to examine and derive insights related to FNSV supply and on the effects of the COVID-19 pandemic on food availability. Ethical approval was received from the University of Wollongong Human Research Ethics Committee (2020/246) and CSIRO Social and Interdisciplinary Science Human Research Ethics Committee (187/21).

### 2.3.2. National Food Production Analysis Using FAOSTAT

Food balance sheet FNSV production data for the period of 1961 to 2018 were accessed from the Food and Agriculture Organization of the United Nations' online food and agriculture database FAOSTAT (FAOSTAT (https://www.fao.org/faostat/en/#data (accessed on 28 July 2021)) food balance sheets appended for the years of 1961 to 2013 and 2014 onwards, so noting methodological break between 2013 and 2014). Fruit and vegetable production items in the database were coded as 'other' as opposed to the name of the fruit or the vegetable, so there is no opportunity to understand production by specific crop. Starchy vegetables, spices, and other cash crops were excluded. Data were analysed at the national level and per capita FNSV produced per annum over the period 1961 to 2018.

### 2.3.3. Food Import Analysis Using Pacific Food Trade Database

To characterise imports and exports of FNSV in Solomon Islands, we performed a descriptive quantitative analysis of the Pacific Food Trade Database (PFTD), which is derived from a global trade database BACI HS92, which is sourced entirely from United Nations Comtrade data [44]. Total volume (t) of imports to and exports from Solomon Islands were extracted directly from the PFTD according to the assignment of commodities to each of the four fruit and vegetable groupings. The groups are fresh vegetables, preserved vegetables, fresh and preserved fruit, and legumes (Table A4). The years 1995 to 2018 were chosen to allow analysis of trade data in the context of external shocks to the Solomon Islands food system that occurred during those years, including the civil unrest in that late 1990s. We deleted the 2003 figure for legumes as it was likely an error in country attribution made by the exporter.

### 2.4. Analysis of Policy Actions

A documentary policy content analysis of food systems policies in Solomon Islands was conducted to identify barriers to fruit and vegetable consumption and policy approaches for addressing them. The data source was the policy documents of government sectors relevant to food systems governance, including agriculture, trade, industry, infrastructure, and finance (Table A5). Documents were accessed via government websites, and via colleagues of the author team in Solomon Islands, and these were stored in Nvivo20 for data extraction. Data were extracted from 10 policy documents and analysed against a predetermined coding framework, which drew from Kingdon's concept of 'problem framing' [49] to capture the articulation of food systems 'problems' impacting FNSV acquisition and supply. We mapped the key findings from the HIES, FAOSTAT, and PFTD dataset analysis with the 'problems' as articulated by these policies. We also coded the policy documents for the types of policy solutions being applied across the food system (production, manufacturing, and market access), documenting clear opportunities for sustainable fruit and vegetable consumption.

## 3. Results

### 3.1. Individual Factors and Food Environments

3.1.1. Individual Factors (Economic, Situational) and Food Environment (Affordability) [2012–13 HIES]

Wealthier households were significantly more likely to acquire the minimum daily recommended intake of 400 g FNSV (Table 3). Members of households with a household head aged 40 and above, members of households that participate in fishing, members of households with seven or more members, and members of households in urban areas had significantly lower likelihood of acquiring the daily recommended amount of FNSV.

**Table 3.** Associations between sociodemographic grouping and whether household members meet minimum requirement for 400 g fruit or non-starchy vegetables per day (source: 2012–13 HIES).

| Sociodemographic Factor (Referent Category in Brackets) † | Does Acquire 400 g FNSV per Day vs. Does Not | |
|---|---|---|
| | **Odds Ratio (95% CI)** | ***p* Value** |
| Household head aged 40+ (15–39) | **0.752 (0.588 to 0.961)** | **0.023** |
| Household participates in fishing (Does not participate) | **0.682 (0.545 to 0.853)** | **0.001** |
| Household has 7 or more members (0–6) | **0.439 (0.350 to 0.549)** | **<0.001** |
| Higher 50% wealth households (Lower 50% wealth households) | **2.311 (1.920 to 2.782)** | **<0.001** |
| Urban (Rural) | **0.531 (0.439 to 0.642)** | **<0.001** |
| Household head has tertiary education (Junior secondary) | 0.995 (0.800 to 1.237) | 0.965 |

† Marital status of household head, sex of household head, household has a wage income, household has a vegetable garden status, household participates in agriculture status are not presented here because they produced *p* values greater than 0.1 in the bivariate analysis step (Step 1 in explanation of multivariate analysis steps in Methods—see Table A2). Results in non-bold indicate results of Step 2 that returned *p* values > 0.1 (Table A3). These were not included in Step 3 (the best fit model), the results of which are presented in bold above.

Almost all households consumed some FNSV in Solomon Islands (Figure 1). Household expenditure on FNSV was less than that required for 400 g of FNSV/day among 60% of households; one quarter of households spent less than half of the necessary expenditure. The cash value (price) of 400 g/person/day FNSV in urban areas was SBD 2.48, and in rural areas SBD 1.62. In urban areas, three-quarters of FNSV is sourced via cash purchases, while in rural households, 83 percent is from home production (Table A6).

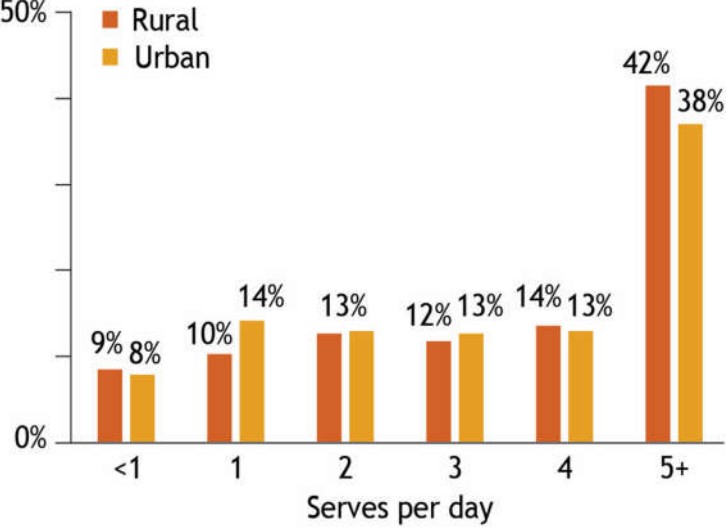

**Figure 1.** FNSV acquisition: Percentage of households acquiring specified serve per person (source: 2012–13 HIES).

3.1.2. Food Environments (Food Availability and Vendor Properties) [2012–13 HIES]

At the national level, just under three-quarters of FNSV was acquired from gardens and plantations, however there were significant differences between rural and urban locations. In rural areas, the majority of FNSV was sourced by consumers directly from the natural food environment, whereas in urban areas the retail food environment was the main source—predominantly central food markets (Table 4).

**Table 4.** Proportion of total quantity of household fruit and vegetable acquisition by food environment type and sub-type [47].

| | Proportion of Total Quantity of Fruit and Vegetables Acquired by Food Environment Type | | | | | | | | | | | | |
|---|---|---|---|---|---|---|---|---|---|---|---|---|---|
| | **Location** | | | **Province (City)** | | | | | | | | | |
| | **National** | **Urban** | **Rural** | **Choiseul** | **Western** | **Isabel** | **Central** | **Rennell and Bellona** | **Guadalcanal** | **(Honiara)** | **Malaita** | **Makira** | **Temotu** |
| Formal retail | 5.3% | 40% | 1.6% | 1.5% | 1.9% | 0.1% | 1.0% | 0.0% | 6.5% | 54% | 2.3% | 0.1% | 0.4% |
| Informal retail | 6.6% | 34% | 3.6% | 5.9% | 8.5% | 1.5% | 4.9% | 0.5% | 5.4% | 33% | 5.6% | 2.1% | 4.2% |
| Cultivated | 71% | 18% | 76% | 78% | 74% | 90% | 72% | 86% | 72% | 7.7% | 68% | 85% | 71% |
| Wild | 11% | 1.1% | 12% | 5.0% | 6.6% | 5.5% | 13% | 10% | 13% | 0.3% | 18% | 4.4% | 16% |
| Kin and community | 5.8% | 6.0% | 5.8% | 9.2% | 7.1% | 3.3% | 7.4% | 3.6% | 3.3% | 5.1% | 5.4% | 7.5% | 8.4% |
| Not coded | 0.6% | 0.7% | 0.6% | 0.8% | 1.8% | 0.0% | 1.4% | 0.2% | 0.1% | 0.1% | 0.6% | 0.6% | 0.3% |
| Column total | 100% | 100% | 100% | 100% | 100% | 100% | 100% | 100% | 100% | 100% | 100% | 100% | 100% |
| | Proportion of Total Quantity of Fruit and Vegetables Acquired by Food Environment Sub-Type | | | | | | | | | | | | |
| | **Location** | | | **Province (City)** | | | | | | | | | |
| | **National** | **Urban** | **Rural** | **Choiseul** | **Western** | **Isabel** | **Central** | **Rennell and Bellona** | **Guadalcanal** | **(Honiara)** | **Malaita** | **Makira** | **Temotu** |
| Stores & shops | 0.5% | 2.5% | 0.3% | 0.6% | 0.7% | 0.1% | 0.3% | 0.0% | 0.9% | 2.3% | 0.2% | 0.1% | 0.1% |
| Supermarkets | 0.0% | 0.1% | 0.0% | 0.0% | 0.0% | 0.0% | 0.0% | 0.0% | 0.0% | 0.2% | 0.0% | 0.0% | 0.0% |
| Restaurants | 0.0% | 0.2% | 0.0% | 0.0% | 0.0% | 0.0% | 0.0% | 0.0% | 0.1% | 0.2% | 0.0% | 0.0% | 0.0% |
| Co-operatives | 0.0% | 0.0% | 0.0% | 0.0% | 0.0% | 0.0% | 0.5% | 0.0% | 0.0% | 0.0% | 0.0% | 0.0% | 0.0% |
| Central markets | 4.7% | 37% | 1.2% | 0.9% | 1.2% | 0.0% | 0.2% | 0.0% | 5.5% | 51.2% | 2.1% | 0.0% | 0.3% |
| Local markets | 5.4% | 31% | 2.6% | 5.4% | 7.3% | 1.3% | 4.3% | 0.2% | 4.3% | 29.0% | 4.3% | 1.4% | 2.1% |
| Canteen | 0.3% | 1.0% | 0.3% | 0.1% | 0.2% | 0.1% | 0.2% | 0.1% | 0.3% | 0.9% | 0.3% | 0.3% | 0.8% |
| Opportunistic vendors | 0.9% | 2.2% | 0.7% | 0.4% | 0.9% | 0.1% | 0.4% | 0.3% | 0.8% | 3.1% | 0.9% | 0.5% | 1.4% |
| Gardens | 44% | 17% | 47% | 49% | 50% | 62% | 47% | 45% | 45% | 7.6% | 39% | 53% | 38% |

**Table 4.** *Cont.*

| | Location | | | Province (City) | | | | | | | | | |
|---|---|---|---|---|---|---|---|---|---|---|---|---|---|
| | **National** | **Urban** | **Rural** | **Choiseul** | **Western** | **Isabel** | **Central** | **Rennell and Bellona** | **Guadalcanal** | **(Honiara)** | **Malaita** | **Makira** | **Temotu** |
| Plantations | 27% | 1.9% | 29% | 29% | 24% | 28% | 25% | 41% | 27% | 0% | 29% | 32% | 32% |
| Bush | 9.4% | 0.8% | 10% | 3.2% | 5.1% | 4.6% | 12% | 10% | 12% | 0% | 15% | 4.0% | 15% |
| Ocean | 1.0% | 0.1% | 1.1% | 0.4% | 0.6% | 0.2% | 0.9% | 0.0% | 0.0% | 0.1% | 3.0% | 0.2% | 0.9% |
| Rivers | 0.3% | 0.0% | 0.3% | 1.0% | 0.5% | 0.3% | 0.1% | 0.1% | 0.5% | 0.0% | 0.1% | 0.2% | 0.1% |
| Estuary | 0.3% | 0.1% | 0.3% | 0.6% | 0.4% | 0.4% | 0.7% | 0.1% | 0.0% | 0.0% | 0.4% | 0.0% | 0.3% |
| Family and community | 5.6% | 5.8% | 5.6% | 9.0% | 7.1% | 3.1% | 7.3% | 3.6% | 3.3% | 5.0% | 5.3% | 7.2% | 8.1% |
| Cultural and social gatherings | 0.1% | 0.1% | 0.1% | 0.2% | 0.0% | 0.1% | 0.1% | 0.0% | 0.1% | 0.0% | 0.2% | 0.3% | 0.4% |
| Not coded | 0.6% | 0.7% | 0.6% | 0.8% | 1.8% | 0.0% | 1.4% | 0.2% | 0.1% | 0.1% | 0.6% | 0.6% | 0.3% |
| Column total | 100% | 100% | 100% | 100% | 100% | 100% | 100% | 100% | 100% | 100% | 100% | 100% | 100% |

In terms of diversity of types of FNSV available to households, banana, papaya, watermelon, and pineapple accounted for 75 percent of fruit, while island cabbage and other cabbages made up 60 percent of non-starchy vegetables, with tomato, green beans, and eggplant accounting for another 20 percent.

*3.2. Food Supply Chains*

3.2.1. Local Markets and Production Analysis Using Vendor Survey of Urban Food Markets

Sourcing of produce for sale.

The vast majority of produce sold by market vendors in in Auki and Gizo originated from within the province of sale (Malaita and Western Province, respectively). Produce sold at Honiara Central Market mostly originated from the same province (Guadalcanal), but produce from across the other provinces was also sold in Honiara. Most produce sold at open markets was sourced from the vendors' own garden or plantation, or harvested from the wild by the vendor themselves. A smaller portion of vendors purchased FNSV from producers (usually via cash) to then be sold—this was most common in Honiara, but was still the case for less than one-third of produce sold.

Various issues related to the natural environment were reported by market vendors in relation to producing, sourcing, and selling produce at markets, with the most commonly cited issues in all locations being weather, natural disasters, and pests (Giant African Snail, and a variety of other invertebrate species spoiling crops). In Honiara, the most common issue mentioned in relation to weather and disasters was flooding.

Market response to COVID-19 (external shock to food system).

The COVID-19 pandemic caused notable changes in market produce accessibility, price, and waste. Across all three locations, market vendors reported more difficulties making financial 'ends meet' since the COVID-19 pandemic had started: a major change reported by respondents in 2020 was reduced cash flow and customers in the market. Market vendors in Honiara in particular often reported increased wastage of produce in relation to lockdowns and closures.

The perceived impact of a recent government policy to encourage members of the urban population to return to their home province (usually villages in rural areas) on local food systems varied, as relocation was for varying periods of time. Many respondents noted the pressure their own household had been under to feed extra people, either due to job losses or relocation of family members between urban and rural areas.

3.2.2. National Food Production Analysis Using FAOSTAT

Total FNSV production increased over the period of 1961 to 2018, with fruit production accounting for the majority of increase in production over the period (Figure 2). However, on a per capita basis, annual FNSV production is insufficient to supply the population with the recommended 400 g per person per day. Per capita FNSV production has been declining since 1960 where production levels have reduced from around 220 g per capita per day to around 160 g per capita per day in 2018.

3.2.3. Food Import Analysis Using Pacific Food Trade Database

The combined total of imported FNSV provided a trivial proportion of the 400 g/person/day recommended for good health. Exports of FNSV from Solomon Islands were negligible (Figure 3). Key FNSV that are imported include alliaceous vegetables (e.g., onions, garlic) which made up around two-thirds the annual average in 2014 to 2018, followed by apples, and non-starchy root vegetables, fresh oranges, and preserved vegetables.

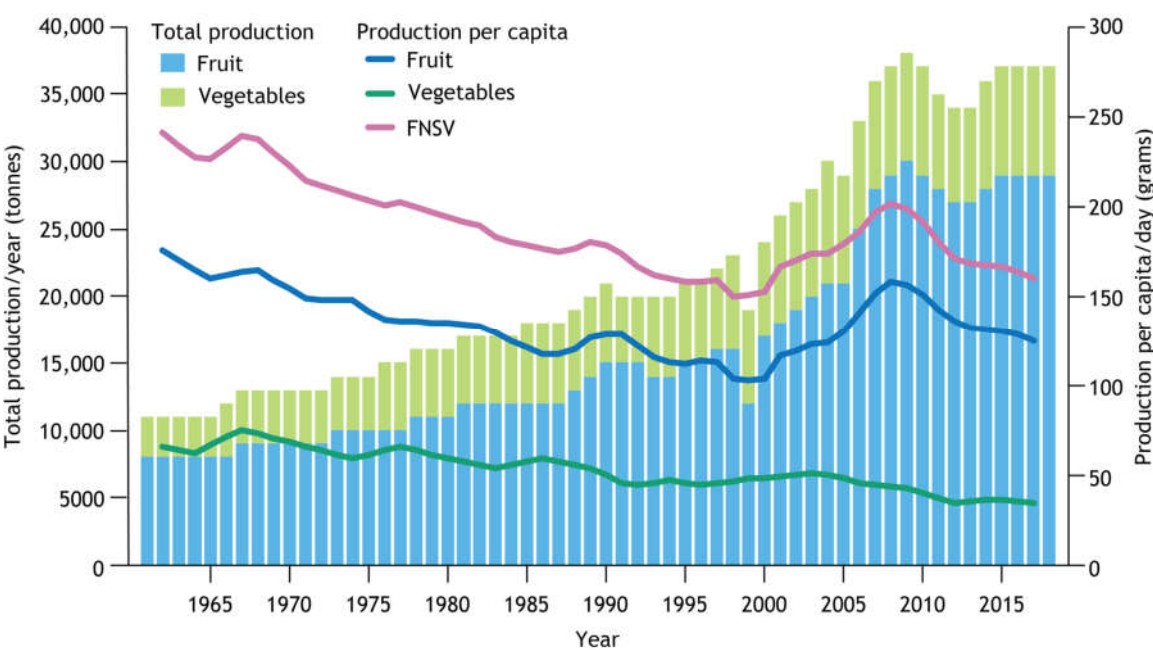

**Figure 2.** National level FNSV production (source: FAOSTAT).

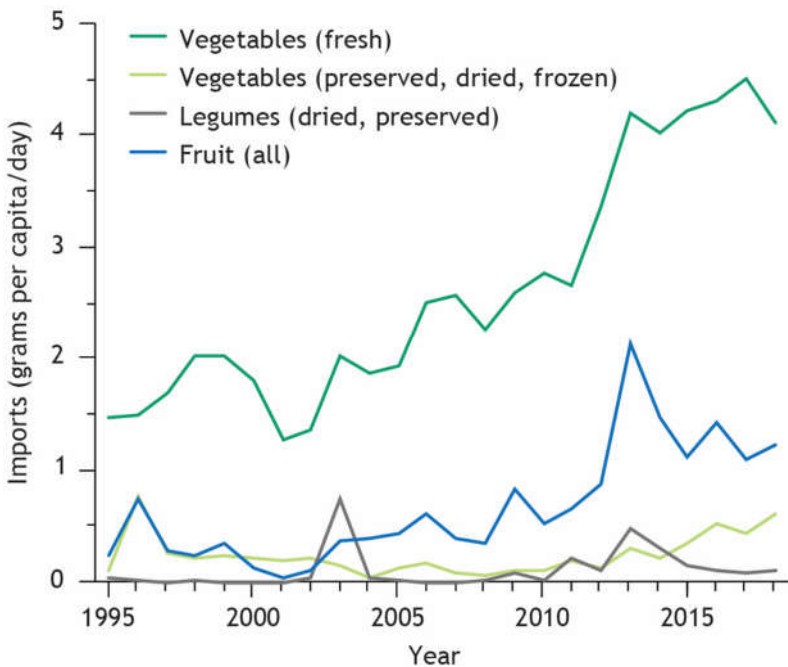

**Figure 3.** Imports of fruit and non-starchy vegetables to Solomon Islands (grams per capita per day) 1995–2018 (source: The Pacific Food Trade Database).

### 3.3. Barriers to FNSV Consumption

We identified several key issues related to the provision and consumption of FNSV through our analysis of the HIES, FAOSTAT, and PFTD and matched these to the way they were framed as problems in the Solomon Islands Government's food system-relevant policy (Table 5). In particular, barriers to supply and consumption were framed around the need to improve market infrastructure, the need for technical capacity for FNSV production, and the need for investment in small and medium sized enterprises.

**Table 5.** Key findings from HIES, FAOSTAT and PFTD analysis mapped to policy.

| Food System Component ^ | Key Findings | Data Source | Policy Analysis Problem Framing * |
|---|---|---|---|
| Individual factors | Those in urban areas have significantly lower odds of acquiring the daily recommended amount of FNSV than rural areas | HIES (multivariate analysis) | |
| Individual factors | More than half of households do not acquire sufficient FNSV. The following have significantly lower odds of acquiring the daily recommended amount FNSV (400 g): <br>• People with household head aged 40 and above, <br>• People with seven or more household members, <br>• People who live in households that participate in fishing <br>• Urban areas <br>• Lower income households | HIES (multivariate analysis) | • Constraints around land ownership and access, limiting opportunities for youths, entrepreneurs, and investors in agriculture, particularly in urban areas; Limited transport and infrastructure for providing access to markets (Solomon Islands Agriculture Sector Growth & Investment Plan 2021–2030, Solomon Islands Ministry of Agriculture and Livestock) <br>• Roads in poor state hinder trade and access to local markets. Recurrent budget for government to maintain all roads and to build new ones is inadequate (National Infrastructure Investment Plan (2013), Ministry of Development Planning and Aid Coordination) <br>• Limited private sector development, with high failure rate for small to medium enterprises that might pursue food processing, preserving and manufacturer; Limited access to finance and investment; Limited product knowledge or export market opportunities (Solomon Islands Micro, Small, and Medium Enterprises Policy and Strategy, 2012, Ministry of Commerce, Labour, Industries & Immigration) <br>• Limited ability to effectively deliver resources and services to farmers; Farming knowledge and access to 'improved' methods and technologies (Solomon Islands Agriculture Sector Growth & Investment Plan 2021–2030, Solomon Islands Ministry of Agriculture and Livestock) <br>• Knowledge on manufacturing, harvesting and post-harvesting food processing techniques, food handling and storage techniques is limited; Post-harvest losses, associated with inadequate market facilities and storage, and access to facilities to pursue food processing, preserving, and manufacturing. (Solomon Islands Agriculture Sector Growth & Investment Plan 2021–2030, Solomon Islands Ministry of Agriculture and Livestock) |
| Food environments | Cash value (price) of 400 g/person/day FNSV <br>urban areas: SBD 2.48 <br>Rural areas: SBD 1.62 | HIES | |
| Food environments | FNSV consumed in rural areas were 82% (fruit) and 84% (non-starchy vegetables) home produced. Urban areas: fruit and non-starchy veg were acquired 66% (fruit) and 80% (non-starchy vegetables) via cash (i.e., purchased), respectively. | HIES | |
| Food environments | In rural areas, majority of FNSV acquired from cultivated (76%) and wild (12%) food environments. In urban areas, majority of FNSV acquired from formal retail (40%, and central markets made up 37%) and informal retail (34%, and local markets supplied 31%). | HIES | |

**Table 5.** *Cont.*

| Food System Component ^ | Key Findings | Data Source | Policy Analysis Problem Framing * |
|---|---|---|---|
| Food supply chains | Vendors reported notable impacts of weather events, especially flooding, on their produce supply in recent years | Market vendor survey | • Vulnerability of transport and ICT infrastructure to climate risks and disasters (National Infrastructure Investment Plan (2013–2023), Ministry of Development Planning and Aid Coordination) |
| Food supply chains | There was higher reported produce waste in Honiara than in Auki or Gizo | Market survey | • No organic waste collection from markets for composting and organic fertilization (Solomon Islands Agriculture Sector Growth & Investment Plan 2021–2030, Solomon Islands Ministry of Agriculture and Livestock)<br>• Unreliable access to urban and export markets (Solomon Islands Trade Policy Framework (2015)<br>• Lack of waste management and recycling facilities for private or public sector, lack of an overall solid waste policy and legislative framework (National Infrastructure Investment Plan (2013–2023), Ministry of Development Planning and Aid Coordination)<br>• A need to improve market infrastructure including storage facilities so that vendors can store food (Solomon Islands Agriculture Sector Growth & Investment Plan 2021–2030, Solomon Islands Ministry of Agriculture and Livestock) |
| Food supply chains | Pests, e.g., Giant African snails | Market survey | • Vulnerability of agricultural production to climate-related extremes, pests, and disasters (Solomon Islands Agriculture Sector Growth & Investment Plan 2021–2030, Solomon Islands Ministry of Agriculture and Livestock)<br>• Unsustainable exploitation of natural resources (e.g., logging) (Solomon Islands Trade Policy Framework (2015), Solomon Islands Government)<br>• Improper use and storage of hazardous pesticides (Solomon Islands Agriculture Sector Growth & Investment Plan 2021–2030, Solomon Islands Ministry of Agriculture and Livestock) |
| Food supply chains | Most produce sold at open markets is sourced from the vendors' own garden or plantation, or harvested from the wild by the vendor themselves. A smaller portion of vendors purchase FNSV from producers (usually via cash) to then be sold—this is most common in Honiara, but is still less than one-third of produce sold. | Market survey | • A need to establish cooperatives for the buying and selling of fruit, apart from those for export market (e.g., Noni, Coconut) [50] |

**Table 5.** *Cont.*

| Food System Component ˆ | Key Findings | Data Source | Policy Analysis Problem Framing * |
|---|---|---|---|
| Food supply chains | Production of FNSV is increasing, but on a per capita basis it is declining. Solomon Islands is currently not producing enough for each person to consume the recommended 5 serves FNSV per day. | FAOSTAT | • Policy gap in some policies in defining this issue as a problem, e.g., "Subsistence food crop production represents a major strength of the Solomon Islands economy and food production has kept pace with population growth." (SIG National Development Strategy 2016–2035) <br> • Limited capacity of the agricultural sector to offer effective advisory services and to coordinate between public and private partners; Limited transport and information communication technology (ICT) infrastructure for providing access to markets, and to effectively deliver resources and services to farmer. Farming knowledge and access to 'improved' methods and technologies (Solomon Islands Agriculture Sector Growth & Investment Plan 2021–2030, Solomon Islands Ministry of Agriculture and Livestock) |
| Food supply chains | Imported FNSV provide a negligible proportion of the 400 g/person/day recommended for good health | Pacific Food Trade Database | • The Trade Policy Framework includes specific mention of fruit and vegetables, and highlights challenges related to domestic production (including agricultural production, biosecurity, and value adding). |

ˆ Food system components from food system framework in Figure 1 in Fanzo et al. 2021 [2]. * We note there is cross-over between some policies, and some points are relevant to multiple findings from our data analysis.

### 3.4. Analysis of Policy: Solutions Framing

A number of options to facilitate increased production, distribution, and consumption of FNSV were identified in the policy documents analysed in this study. These included investment in local and environmentally sustainable production, and efforts to increase livelihood equity through targeting youth and women (Table 6).

**Table 6.** Potential opportunities identified in existing policy to increase FNSV production and supply.

| Aspect of Food Systems | Policy Approaches Identified | Policy |
|---|---|---|
| Promoting sustainable production of fruits and vegetables | • Technological development and advancement for producers and manufacturers to promote efficient, sustainable farming systems<br>• Promotion and training on cultivation, nutritional value and usage of fruit and vegetables in collaboration with Health<br>• On-farm trials for exotic highland vegetable production, underpinned by principles around resilience and market demand | Solomon Islands Agriculture Sector Growth & Investment Plan 2021–2030, Solomon Islands Ministry of Agriculture and Livestock |
| | • Research into traditional practices of production and preservation<br>• Conservation, multiplication and distribution via seed preservation and seedlings | Solomon Islands Indigenous Fruit and Nut Industry Policies and Strategies 2014–2020 |
| | • Amend or update forestry laws to assist land owning groups including farmers to promote establishment of forest plantations including reforestation and promotion of sustainable harvesting | Solomon Islands National Development Strategy 2016–2035 |
| | • Non-discriminatory access to rural land for productive purposes for foreign investors | Solomon Islands Trade Policy Framework (2015), Solomon Islands Government |
| | • Develop agriculture and livestock through agricultural marketing and land use planning to improve food security, livelihoods, and community sufficiency in rural areas; use targeted multi-disciplinary interventions to diversify agriculture and promote agribusiness and alternative livelihoods | Solomon Islands National Development Strategy 2016–2035 |
| | • Conduct research and propose livestock feed formulations (especially for poultry and pigs) which are based on locally available raw material (e.g., palm oil, copra meal, fish meal, cassava, sweet potato) as much as possible | Solomon Islands Agriculture Sector Growth & Investment Plan 2021–2030, Solomon Islands Ministry of Agriculture and Livestock |
| Scaling up fruit and vegetable production and processing | • Research and capacity building in technological enhancements to harvesting and post-harvesting food handling and storage, for instance hydroponics, seaweed fertilisation | Solomon Islands Agriculture Sector Growth & Investment Plan 2021–2030, Solomon Islands Ministry of Agriculture and Livestock |
| | • Loans and equipment to support entrepreneurs with food processing and manufacture, particularly for women and youths<br>• Engagement of youth in small-medium enterprises specific to agro-processing or agro-service provision<br>• Training in business development and business incubation | Corporate Plan (2020–2024), Ministry of Commerce, Industry, Labour & Immigration |
| | • Capacity building of private sector and entrepreneurs on high value production, processing, and marketing | Solomon Islands Trade Policy Framework (2015), Solomon Islands Government |

**Table 6.** *Cont.*

| Aspect of Food Systems | Policy Approaches Identified | Policy |
|---|---|---|
| | • Promote and enhance sustainable subsistence-based farming systems including organic farming, indigenous crops and improved post-harvest handling<br>• Consider incentives and possibly the reintroduction of initial "development" subsidies for expanding cash crop production and local agriculture food gardens to go into mass production. | Solomon Islands National Development Strategy 2016–2035 |
| | • National and provincial investment in transport, market infrastructure, warehousing and cold storage, electricity, and mobile phone coverage<br>• Building productive relationships to promote coordination and linkages across the food chain between producers, processers, and traders. | Solomon Islands Agriculture Sector Growth & Investment Plan 2021–2030, Solomon Islands Ministry of Agriculture and Livestock |
| | • Capacity building of private sector and entrepreneurs on high value production, processing, and marketing<br>• Creation of 'backwards linkage' programs to connect farmers as local suppliers to mining, fishing and hospitality industries | Solomon Islands Trade Policy Framework (2015), Solomon Islands Government |
| Proving greater access to domestic markets | • Facilitate and provide incentives to financial institutions and credit unions to cater and provide financial and soft loans to small and medium enterprises (SMEs) and rural communities.<br>• Strengthen the environment for the development of SMEs including policies and incentive packages in targeted growth sectors<br>• Review viable options to improve access to rural financial services for savings and credit including options for: (i) mobile/telephone banking and micro-finance; (ii) rural people's bank to provide credit and financial services to local business; and (iii) special rural financing schemes. Improve the quality and range of financial services to support private sector development.<br>• Provide support and foster an enabling environment for investment for young entrepreneurs to venture into potential identified industries<br>• Develop programmes for rural areas to encourage growing of local crops through the creation of local markets. Where appropriate, introduce new sustainable crops.<br>• Build road access to interior areas with agriculture potential to enable population to access their own land for agriculture development. | Solomon Islands National Development Strategy 2016–2035 |

## 4. Discussion

In this paper, we analysed food systems data from a diverse range of sources and combined this with policy analysis to identify barriers to, and opportunities for, increased consumption in Solomon Islands. Key findings related to supply were the findings that domestic production of FNSV is insufficient, while per capita national level supply through imports is inconsequential. The food environment analysis indicated multiple further issues hampering FNSV consumption, and the three most important insights were: (1) those in urban areas are less likely to acquire sufficient FNSV for good health than those in rural areas, (2) there is a large difference between the source of acquisition by location; in rural areas FNSV are mostly acquired from home cultivation, while in urban areas, FNSV are mostly purchased from markets, and (3) combined results of the multivariate analysis and pricing analysis revealed urban affordability is an important access lever of FNSV.

The policy analysis showed a range of strengths In the current policy environment in Solomon Islands. In particular, we found explicit priorities in policy documents for investment in local production of FNSV and efforts to increase livelihood equity through targeting youth and women. We also saw a recognition of the need for the food systems sectors to address sustainable and equitable consumption and supply of fruit and vegetables by adopting a broader range of policy instruments. This demonstrates a shift away from a formerly 'productivist' attitude towards consideration of a broader food system

approach [51]. In order to address consumption deficits however, Solomon Islands needs to consider the resources required to operationalise local-level action and the implementation of enhanced production systems [15,51,52].

There are myriad interconnected ways FNSV supply and acquisition can influence SDGs [15,17,40,52–54] in Solomon Islands. These are summarised in Figure 4 below.

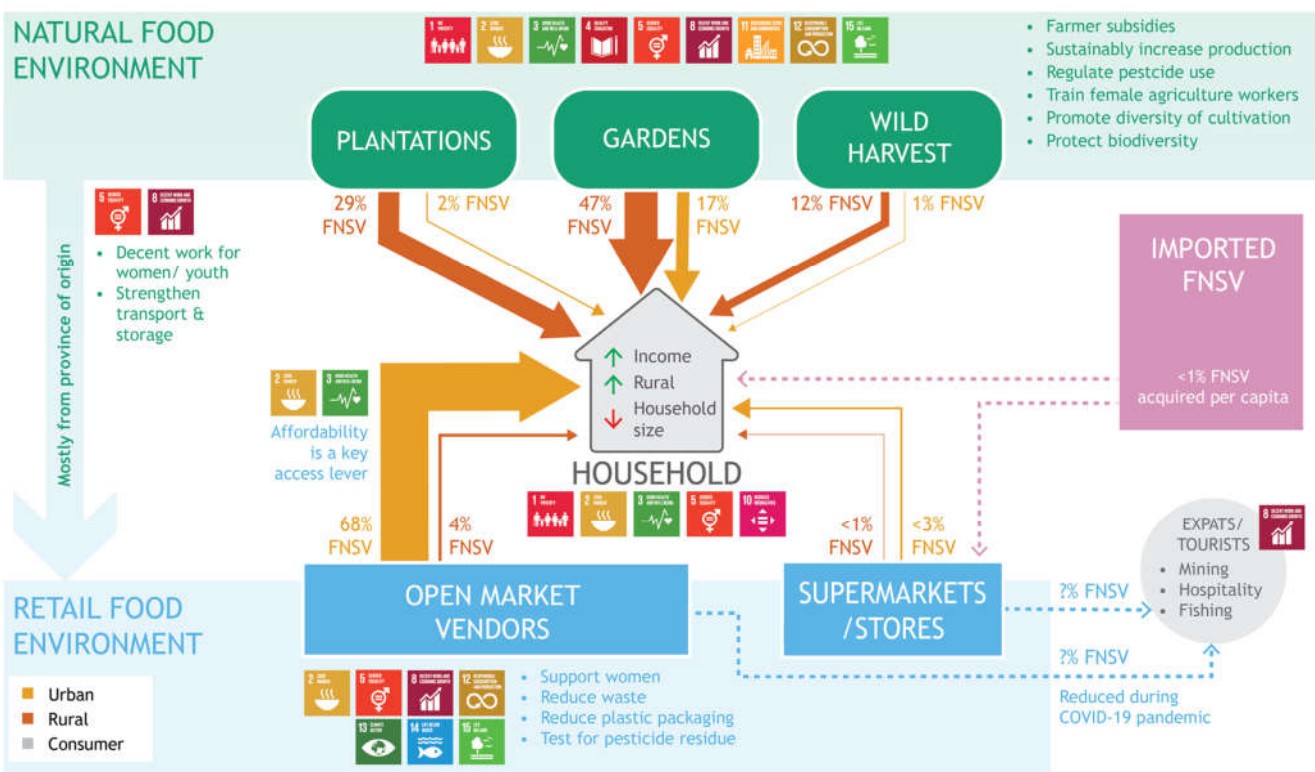

**Figure 4.** FNSV acquisition flow and key opportunities to optimise food system outcomes.

Here, we extrapolate on our analysis of Solomon Islands government policy through a sustainable food systems lens, to present key opportunities for multiple gains across the food system components of the (1) individual consumption from food environments, (2) supply and markets, and (3) production. Resilience to shocks including climate change, civil unrest, and pandemics are cross-cutting areas [2], which need to be interwoven into each of these components.

## 5. Affordability

Fruit and vegetable affordability is an important barrier to consumption (SDGs 2 and 3), in particular for urban populations, large households, those on low incomes, and those dependent on formal retail. Our analysis showed that people living in urban areas are less likely to consume the daily recommended amount of FNSV than those in rural areas. There is likely a complex interplay of food environment and individual factors contributing to this phenomenon including transport, lower access to gardens, convenience, individual taste preference, and limited to no cold storage for fresh foods for households and at markets. However, affordability is a key issue in need of policy attention: firstly, our multivariate analysis showed that when all variables are accounted for, household income is a key access lever for FNSV acquisition. Second, the relative cash value of FNSV was higher in urban areas, which suggests that FNSV affordability is a key challenge faced by urban dwellers, in particular those with low incomes. Third, in urban areas, FNSV are predominantly acquired via purchase from market vendors. In Honiara, the main urban centre of Solomon Islands, over half of FNSV are sourced from formal retail and another one-third from informal retail. This finding is important in the context of previous research

which has shown that households that are reliant on formal retail, which includes 'central markets' (open air fresh food markets), acquire less FNSV [47]. The importance of price is supported by previous research in Solomon Islands, which showed that FNSV acquisition in urban areas is influenced by price—this is a problem that appears to have become more prominent over the past decade and a half with the steady rise in the urban population size, and economic disruption related to local and global economic events [21,54].

Intra-household resource distribution is an important factor which appears to be linked with the issue of FNSV affordability. The issue of household size was a resonating theme in our analysis—the multivariate HIES analysis showed that large household size was negatively associated with acquiring sufficient FNSV. An important high level food system driver in recent times is the COVID-19 pandemic and associated measures, and government policies early in the pandemic caused urban populations to fluctuate due to urban-rural migration policies and the effect of travel bans in a somewhat tourism-dependent economy [40]. Our market vendor survey respondents reported issues feeding their own households in this context. In Solomon Islands, the 'Wantok' system means it is a social norm to share resources including food and housing amongst extended kin. The stretching of resources amongst many people can lead to consumption of meals with low nutritional value [21]. Recent research from Vanuatu examined price and affordability and also found that households with more household members were less likely to have sufficient FNSV consumption than households with fewer members [24]. Taken together, our findings suggest that large households and low-income households in urban areas should be a focus for policies and programs aiming to increase FNSV consumption.

Our findings on the importance of affordability to FNSV consumption are consistent with previous research from the Pacific region and worldwide [55,56]. Low FNSV affordability affects the diets of at least 3 billion people globally [34]. A recent Lancet study of 18 countries with varying income categories found that consumption of fruit and vegetables decreased as their cost increased [4]. The 2022 'State of Food Security and Nutrition in the World' report calls for government subsidies for both agricultural production and consumer-level accessibility of nutritious food—which is currently less supported than staple foods such as rice and animal-source foods worldwide [57]. Similarly, a recent report from the FAO highlights the importance of affordability in increasing consumption of fruit and vegetables in sustainable food systems: "Transforming the fruit and vegetable sector requires a holistic and comprehensive approach that looks at the interconnectivity between demand, supply, socio-economic development and fruit and vegetable prices—a major determinant of consumer behaviour" [3]. In Solomon Islands, vegetables and fruits are, respectively, seven and four times more expensive per kilocalorie than roots, tubers, plantains and their products, however, it is critical to note with such comparisons that non-starchy vegetables are relatively low calorie compared with starchy vegetables, but are nutrient dense [18].

Globally, countries are considering pricing policies that can address affordability. Though subsidies on fruit and vegetables have proven impactful in other settings [58], these are not suited to highly informal food environments that are not usually subject to tax. Given this, a key insight for policy makers is that in settings like Solomon Islands, approaches to address price and affordability need to extend across the whole food system, targeting all aspects from production and harvest (or import) through to acquisition from the food environment. Policy approaches in need of focussed consideration include: (a) financial incentives for farmers to produce FNSV for local consumption rather than cash crops for export [production level]; (b) practical support to market vendors for storage and transport to minimise loss and waste of produce [supply chains] [50,59]; (c) additional business support for small and medium sized enterprises (SMEs) including market vendors in unprecedented situations—for instance recent rapid decrease in market demand related to the COVID-19 pandemic and severe disruptions to business during civil unrest [supply chains]; (d) policies that address relative affordability for the consumer—for instance taxes for unhealthy foods and sugar-sweetened beverages which have been recommended in the

Multisectoral National Non-Communicable Disease Strategic Plan 2019–2023 [import and supply chains] [59]; and (e) trialling of vouchers that can be used to purchase FNSV, especially for those identified as vulnerable to low consumption in our analysis, for example those on low incomes, living on large households, in households with no access to land to grow FNSV, and the elderly and disabled [consumer level]. Poverty reduction in general has an important role to play in access to adequate nutritious food [23].

## 6. Equitable and Inclusive Supply Chains

Investing in equitable fruit and vegetable supply processes will contribute not only to nutrition, but also to livelihoods and poverty reduction. Fresh food markets are critical to FNSV supply, particularly in urban areas where our analysis showed they provide over two-thirds of the FNSV consumed. Marketing (sale of FNSV at open air fresh food markets) is a significant contributor to the Solomon Island economy. For example, the annual turnover at the Honiara Central Market is estimated to be between 10 and 16 million USD [60] and plays a vital role in forming linkages between rural and urban income generation, rural food production, household food and nutrition security, and economic activity related to transportation and reselling [22], and unquantified social and cultural connection. However, in order to maximise the economic co-benefits and progress across the SDGs, there is a need to prioritise equity across the FNSV supply chain with a focus on inclusive and decent work opportunities for women and youth (SDG 8) at all steps.

In Solomon Islands, women make up 80–90% of fresh FNSV market vendors [50,59,60]. Market vendors encounter numerous difficulties: often transport routes to market are long and difficult, [22] women sometimes stay overnight at markets [41] raising potential safety concerns, and formal market licensing arrangements can be unreliable and difficult to navigate [60]. Improvements in market transportation and storage are needed [22]. There have been recent policy commitments in Solomon Islands to improve these conditions and opportunities, including market infrastructure upgrades to develop "more convenient and safe market places" [59]. Economic empowerment, household food security, and women's autonomy in households tends to occur hand-in-hand; there is worldwide evidence that women's autonomy within a household is positively linked with their household's nutritional status [61]. In Solomon Islands, there are inequalities in women's access to resources and control of decision making within households, and disputes on these issues have been linked with domestic violence, which affects two-thirds of women in Solomon Islands [62]. Improving and protecting women's economic empowerment is likely to have important flow-on effects for SDG 5 across multiple targets, including targets 5.2 (eliminate all forms of violence against women) and 5.4 (recognise and value unpaid care and domestic work).

A key point for food systems policy here is that commitments to offer grants, processing equipment, and business training in agriculture could be prioritised towards creating opportunities for women. One example of this in action is the fact that the Ministry of Agriculture and Livestock agricultural extension services are training female-extension officers at the community level. This will open opportunities for women to engage in entrepreneurial activity related to local food production, and help to develop an understanding of what women need at the household level to feed their families [50].

## 7. Waste

Overall in Solomon Islands, the data on FNSV waste across the supply chain reports relatively low wastage [59,63], although more research is needed in this area. However, there are indications of room for improvement, especially in the area of storage at markets, and when there are sudden changes to consumer demand (e.g., related to COVID-19) [40,50] which was reflected by market vendors in Honiara in our market survey. Longer supply routes potentially increase the risk of waste due to post-harvest time prior to consumption, and this was seen in Honiara in our market survey and also by Iese (2021) [40], but not by Underhill (2019) [63]; this could have been linked to the type of produce sold by vendors in the different studies.

We saw in our policy analysis the government committing to a range of strategies to reduce waste, including by supporting widespread adoption of composting systems and by addressing market infrastructure. This has potential important implications for women's market access and vendor safety (SDG 5.2, SDG 8): women often spend many hours in transit carrying heavy produce to market, which is wasted if it is not sold. Secure cold storage may allow vendors to leave the market at the end of the day rather than sleeping on market floors.

In addition to waste minimisation, the impact of marketing on the natural environment can be further minimised by getting rid of single use plastics from sale processes (SDGs 12, 14). Plastic pollution is an environmental issue of high concern globally [64]. Other Pacific countries have now banned single-use plastics, in many cases substituting with locally woven materials. This level of commitment and innovation can serve as an example for the other Pacific nations and beyond [65].

## 8. Food Production

In-country FNSV production is effectively the only source of FNSV available to the Solomon Islands population and needs to be increased to support the health of the nation and progress on SDGs 2 and 3. The decline in per capita in-country FNSV production represents a missed opportunity for locally produced FNSV to provide essential nutrients to the population, as well as the macroeconomic contribution of FNSV production and the associated labour and value add associated with its supply chain.

FNSV production in Solomon Islands demonstrates many elements of environmentally sustainable practice: supply chains tend to be short as production is predominantly via subsistence farming, packaging is minimal, and production and supply is relatively low waste [59,63]. Although improvements can always be made, these sustainable qualities should be protected and reinforced [40]. FNSV can present a more sustainable alternative to other foods, such as ultra-processed foods, which have been linked to loss of biodiversity [66].

Based on our integrated policy and food systems literature analysis, we identified five policy focus areas to further promote sustainable production of FNSV. The first is to support subsistence farming, including through equitable access to land both in rural and urban areas [59]. Support of subsistence agriculture is well reflected in Solomon Islands' policies, for instance the Solomon Islands National Development Strategy 2016–2035 highlights a need to "Promote and enhance sustainable subsistence-based farming systems". Our analysis showed a need to continue to monitor per capita production and the contribution of subsistence agriculture to this. There is a need to "Promote and enhance sustainable subsistence-based farming systems" (Solomon Islands National Development Strategy 2016–2035). Sustained support will help to advance SDGs 1–10; 12–15 [67].

The second policy area is to support diversity of type of FNSV (SDGs 2 and 3) [3,16]. Our analysis showed that 75% of fruit consumed by households consists of four types of fruit, and 80% of non-starchy vegetables consumed consisted of four types of vegetables. Previous studies in Solomon Islands have raised concern about low levels of FNSV diversity in diets in in Solomon Islands [17,54], and our policy analysis showed that the Solomon Islands Government has recognised this and has plans to scale up introduction of new, exotic, and resilient crops. Consuming a diverse range of fruit and vegetables is important for good nutrition and health (SDGs 2 and 3) [3,16] and a recent study in rural Solomon Islands found an inverse relationship between diet diversity and body fat percentage [10]. A recent review showed that agroecological practices including crop diversity improve food security and nutrition [68]. In terms of consumer preference, there is evidence that if available and accessible, people in Solomon Islands prefer to consume a diverse range of FNSV [54]. Agrobiodiversity is an indicator of food system resilience and stability [2]. A key opportunity we identified in our policy analysis was the need to continue to diversify the types of FNSV produced, including improving the availability of climate change resilient (extreme weather) species. Protecting a diverse range of food environment sources of FNSV

(e.g., sourcing from the wild, imports, and preservation of locally produced FNSV) is also important for food system resilience [17]. This includes protecting the natural environment in Solomon Islands, which has very high biodiversity and hosts a rich Indigenous food system [69].

Third, financial opportunities for producers need to be expanded. Our integrated policy and trade data analysis showed that supporting production of high value crops such as onions [52], which are currently one of the key imported vegetables, would be one approach to meeting the commitment to identifying opportunities to support high value production and livelihoods (Solomon Islands Trade Policy Framework, 2015). Another opportunity is exploring preserved FNSV and legumes—either locally produced or imported. Our trade data show that there are some imports of frozen and preserved vegetables (e.g., frozen broccoli) and legumes occurring, which have high nutritional value. If these were affordable, they could be a useful addition to diets in Solomon Islands. Our policy analysis also revealed an opportunity to support producers to add value to their produce with technological enhancements for post-harvest food handling and processing. Another dynamic and promising policy opportunity within current Solomon Islands policy is the aim to create 'backward linkage' programmes to connect FNSV farmers as local suppliers to mining, fishing, and hospitality industries.

Fourth, for production to be sustainable and good for health, there is a need for pest control and food safety to be aligned—including at the consumer level, for instance though developing capacity to test for chemicals at markets. Safe, regulated, and effective control measures are a key issue in order to ensure supply of FNSV is in fact good for human health and for the environment, and is a known important aspect for consumption behaviour in Solomon Islands [15,52]. This is relevant to SDG 12: responsible production and consumption. Plant pests are a key issue in plant agriculture in Solomon Islands [69]. Increasing local production should not be at the expense of unregulated agrochemical use or non-sustainable land clearing; both issues are in need of sustained policy focus [15,50]. Pests were dominant environmental issue mentioned in our market vendor survey and are the dominant source of FNSV to rural areas. The logging industry is believed to play an important role in the spread of pests such as the Giant African Snail, which was mentioned as a key issue in our market vendor survey [70], and the effects of this external driver on the food system are important for policy makers to consider.

Fifth, the issue of financial resourcing for production of nutritious foods such as FNSV has been highlighted as a global priority, for instance the 2022 FAO 'The State of Food Security and Nutrition in the World' report focusses on repurposing food and agricultural policies and budgets to produce high nutrition foods (e.g., FNSV) to make healthy diets more affordable. This is a complex issue that requires careful consideration of trade-offs on a per-country basis, with careful governance and stakeholder engagement, including SMEs and civil society groups [57]. In Solomon Islands, policy consideration is needed with respect to balancing financial incentives for those in FNSV production reflecting the clear benefits to public health (return on investment) compared with the benefits of cash cropping. It is promising that the Solomon Island agriculture sector has recently made an explicit commitment to protecting land resources for fruit and vegetable production [59].

## 9. Reflections on Food Systems Disruptions and FNSV

There has been a documented trend to turn to local (including peri-urban) production even more in times of crisis—including both the COVID-19 pandemic and natural disasters [40,53]—and also, since imports are so negligible, bolstering FNSV production in Solomon Islands is a critical pathway to resilience of the food system. This is not without complexities and there have been calls to not 'romanticise' local FNSV production in Solomon Islands [17]. The COVID-19 pandemic had mixed effects on local agricultural production and is a good example of the need to consider environmental sustainability and livelihood equity in rapid response food security policy. With the government COVID-19 mitigation policy for people to return from urban areas to rural areas in the early

months of the pandemic, there was a boom in subsistence agriculture, but also unintended consequences including mass land clearing, land disputes and, in some cases, reduced agricultural production due to labour shortages [40]. There was also an increase in food losses and wastage due to overproduction and pests [40]. Vulnerable members of the community affected by labour shortages included single parents, elderly households, and people living in urban informal communities (i.e., those who were not physically able and/or available to cultivate FNSV) [40], which resonates with the finding in our HIES multivariate analysis that households with older household heads were vulnerable to low FNSV consumption. The elderly, along with women and youth mentioned above, are vulnerable community members who need to be supported across the food system, from the level of FNSV production through to consumption.

The issue of diversity of the type of FNSV available for consumption in the context of external shocks to the food system is also important to consider. In many households, diversity of FNSV consumption reduced further in post-COVID-19 times, where urban-rural migration and government emergency food security responses supporting local production actually lowered diversity of agricultural production—and in doing so, decreased resilience of the food supply to climate change and climatic extremes [40]. Diversity in foods available is likely to be vulnerable to other large-scale shocks, for instance our trade data analysis showed there was a notable fall in non-starchy vegetable imports between 1998 and 2001, which aligns with the major civil unrest during this time.

## 10. Strengths and Limitations

This study had several limitations and strengths. We draw on a broad range of data sources to present a multi-dimensional reflection of supply and acquisition of FNSV within the Solomon Island food system, and to identify leverage points for improving health outcomes through increased domestic supply and affordability. Our study harnessed detailed data from the Solomon Islands HIES 2012–2013 dataset, standardised to per-AME and per gram, and compared this across the population. It provides policy relevant information to a previously very constrained area of the food systems literature. As many of the food systems issues in Solomon Islands are replicated in many other countries in the Pacific region, and globally, the food systems approach presented in this paper has the potential to be adapted to other countries.

For the multivariate analysis, in line with previous research, we have interpreted these data using the minimum value of 400 g of fruit and vegetables on a per-capita basis [20,24]. The strength of this approach is that it provides a constant baseline by which to assess household expenditure on FNSV. However, the fact that households are likely to acquire FNSV of varying value (not solely the cheapest), means that we may have overestimated the proportion of the population acquiring 400 g FNSV. We also note that there is a cross-over between categorisation of 'formal' and 'informal' markets in the Pacific food environment typology [47]. There is a history of challenges for women in retaining government licences to sell FNSV at market [60] which could partly explain this cross-over. Fruit and vegetable production items in the FAOSTAT database were all coded as 'other', so there was no opportunity to understand production by specific fruit and vegetable. We did not study traditional systems of governance, and this is an important area for future research. Our policy analysis was limited to documentary review, and future research in this area would likely benefit from detailed stakeholder interviews.

## 11. Conclusions

Analysing food system data in concert with policy analysis allows stakeholders to track progress, as well as enabling policy design to be reflective of changes needed for optimal food system outcomes. In Solomon Islands, local FNSV production needs to be increased sustainably to ensure that the natural environment and food system continues to support the nutrition security of future generations. There is vast opportunity in Solomon Islands for FNSV production and consumption to progress multiple SDGs—especially if

decision makers can ensure economic stimulation, poverty reduction, and food security reduction efforts are aligned with other positive food systems outcomes. To do so will require sustained policy action and multifaceted coordination between agricultural, finance, infrastructure, health sectors and beyond—while navigating the effects of challenges such as climate change and economic disruption.

**Author Contributions:** Conceptualisation, A.M.T. and P.F.; Methodology, A.M.T., P.F., M.K.S., J.R.B., E.R., T.D.B. and A.K.F.; Formal Analysis, P.F., M.K.S., E.R., T.D.B., A.K.F., J.T. and J.R.B.; Data Curation, P.F., M.K.S., E.R., T.D.B., A.K.F., J.T. and S.K.; Writing—Original Draft Preparation, P.F.; Writing—Review and Editing, A.M.T., P.F., M.K.S., E.R., T.D.B., A.K.F. and S.K.; Supervision, A.M.T.; Project Administration, A.M.T., A.K.F.; Funding Acquisition, A.M.T., A.K.F., J.T., M.K.S., T.D.B., J.R.B. and S.K. All authors have read and agreed to the published version of the manuscript.

**Funding:** This study was funded by the Australian Government through ACIAR project FIS-2018-155.

**Institutional Review Board Statement:** The research was conducted in accordance with the Declaration of Helsinki. The market vendor study was approved by the University of Wollongong Human Research Ethics Committee (2020/246 on 14 July 2020) and CSIRO Social and Interdisciplinary Science Human Research Ethics Committee (187/21 on 30 November 2021). The analysis of the existing 2012-13 HIES dataset using the Pacific food system typology categorisations was approved by the CSIRO Social and Interdisciplinary Science Human Research Ethics Committee (035/21 on 11 March 2021).

**Informed Consent Statement:** Informed consent was obtained from all subjects involved in the study.

**Data Availability Statement:** Data available on application to the authors.

**Acknowledgments:** We gratefully acknowledge Douglas Kimie of the Ministry of Finance, Government of Solomon Islands for providing access to the 2012/13 HIES dataset. We also gratefully acknowledge WorldFish Honiara, Anouk Ride, and the vendor survey enumerators for assistance with the market survey, Joel Negin and Mamaru Awoke for advice on the HIES multivariate analysis, Ellen Johnson for dedicated assistance with manuscript preparation, and Eleanor McNeill for graphic design of the figures in this manuscript.

**Conflicts of Interest:** The authors declare no conflict of interest.

## Appendix A

**Table A1.** Descriptive statistics for multivariate analysis.

| | | | Expenditure on FNSV in SBD/AME/Year | | |
|---|---|---|---|---|---|
| | n | Mean | Std. Err | [95% Conf. Interval] | |
| **Age Group** | | | | | |
| 15 to 39 | 1881 | 921.8009 | 35.55722 | 851.3276 | 992.2742 |
| 40+ | 2597 | 713.1713 | 26.73222 | 660.1889 | 766.1537 |
| Sex of household head | | | | | |
| Female | 519 | 958.9041 | 62.48807 | 835.0547 | 1082.753 |
| Male | 3959 | 783.0185 | 30.32907 | 722.9073 | 843.1298 |
| Education group | | | | | |
| Preschool to junior secondary | 2888 | 749.1535 | 32.99557 | 683.7573 | 814.5496 |
| Senior secondary to university | 1318 | 944.3486 | 42.12957 | 860.8491 | 1027.848 |
| Marital status | | | | | |
| Other | 512 | 943.2675 | 58.71104 | 826.9042 | 1059.631 |
| Married | 3966 | 784.4463 | 31.05955 | 722.8873 | 846.0053 |

**Table A1.** *Cont.*

| | | Expenditure on FNSV in SBD/AME/Year | | | |
|---|---|---|---|---|---|
| | n | Mean | Std. Err | [95% Conf. Interval] | |
| Has a wage paying job | | | | | |
| No | 1637 | 752.7717 | 44.07784 | 665.4108 | 840.1325 |
| Yes | 2841 | 831.1309 | 28.89092 | 773.8701 | 888.3918 |
| Participates in agriculture | | | | | |
| No | 1062 | 929.9506 | 48.91778 | 832.9972 | 1026.904 |
| Yes | 3413 | 771.8222 | 33.35853 | 705.7067 | 837.9378 |
| Participates in fishing | | | | | |
| No | 2401 | 912.7687 | 27.7872 | 857.6953 | 967.842 |
| Yes | 2073 | 681.1299 | 44.63374 | 592.6672 | 769.5925 |
| Has a vegetable garden | | | | | |
| No | 760 | 1007.441 | 69.30164 | 870.0869 | 1144.794 |
| Yes | 3718 | 775.4785 | 30.62028 | 714.7901 | 836.1669 |
| Household size | | | | | |
| ≤6 persons | 3016 | 926.9319 | 33.09495 | 861.3388 | 992.5251 |
| ≥7 persons | 1462 | 533.7601 | 19.29735 | 495.5133 | 572.0068 |
| Income 50% | | | | | |
| Below | 2084 | 518.7654 | 20.42201 | 478.2896 | 559.2411 |
| Above | 2394 | 1082.982 | 36.73315 | 1010.178 | 1155.786 |
| Region | | | | | |
| Rural | 3206 | 763.8848 | 34.21588 | 696.0701 | 831.6996 |
| Urban | 1272 | 973.5483 | 38.13829 | 897.9595 | 1049.137 |

**Table A2.** Bivariate analysis results Step 1.

| | Outcome Variable: Does Acquire 400 g FNSV per Day vs. Does Not | |
|---|---|---|
| Input Variables (Referent Category in Brackets) | Odds Ratio (95% CI) | *p* Value |
| Household head aged 40+ (15–39) | 0.642 (0.524–0.786) | 0.000 |
| Household participates in fishing (Does not participate) | 0.735 (0.581–0.929) | 0.010 |
| Household has 7 or more members (0–6) | 0.345 (0.285–0.419) | 0.000 |
| Higher 50% wealth households (Lower 50% wealth households) | 2.502 (2.07–3.019) | 0.000 |
| Urban (Rural) | 0.829 (0.674–1.020) | 0.076 |
| Household head has senior secondary to tertiary education (Junior education) | 1.285 (1.088–1.518) | 0.003 |
| Male household head (Female) | 0.809 (0.604–1.085) | 0.155 |
| Household head married (Not married) | 0.832 (0.640–1.082) | 0.168 |
| Household has a wage income (Household has no wage income) | 1.046 (0.881–1.242) | 0.603 |
| Household has a vegetable garden (Has no garden) | 1.127 (0.901–1.410) | 0.291 |

**Table A3.** Multivariate analysis results Step 2.

| Input Variables (Referent Category in Brackets) | Outcome Variable: Does Acquire 400 g FNSV per Day vs. Does Not | |
|---|---|---|
| | Odds Ratio (95% CI) | *p* Value |
| Household head aged 40+ (15–39) | 0.743 (0.577–0.957) | 0.022 |
| Household participates in fishing (Does not participate) | 0.717 (0.564–0.910) | 0.007 |
| Household has 7 or more members (0–6) | 0.441 (0.347–0.560) | 0.000 |
| Higher 50% wealth households (Lower 50% wealth households) | 2.315 (1.858–2.885) | 0.000 |
| Urban (Rural) | 0.562 (0.459–0.688) | 0.000 |
| Household head has senior secondary to tertiary education (Junior education) | 0.995 (0.800–1.237) | 0.965 |

**Table A4.** Commodities included in trade analysis. HS codes are from the HS92 Coding System.

| HS6 | HS6 Name | Vegetables (Fresh) | Vegetables (Preserved/ Frozen/Dried) | Legumes (Dried/ Preserved) | Fruit (Fresh and Preserved/ Frozen/Dried) |
|---|---|---|---|---|---|
| 70200 | Vegetables: tomatoes, fresh or chilled | 1 | 0 | 0 | 0 |
| 70310 | Vegetables, alliaceous: onions and shallots, fresh or chilled | 1 | 0 | 0 | 0 |
| 70320 | Vegetables, alliaceous: garlic, fresh or chilled | 1 | 0 | 0 | 0 |
| 70390 | Vegetables, alliaceous: leeks and other kinds not elsewhere specified, fresh or chilled | 1 | 0 | 0 | 0 |
| 70410 | Vegetables, brassica: cauliflowers and headed broccoli, fresh or chilled | 1 | 0 | 0 | 0 |
| 70420 | Vegetables, brassica: brussel sprouts, fresh or chilled | 1 | 0 | 0 | 0 |
| 70490 | Vegetables, brassica: edible, fresh or chilled | 1 | 0 | 0 | 0 |
| 70511 | Vegetables: cabbage (head) lettuce (lactuca sativa), fresh or chilled | 1 | 0 | 0 | 0 |
| 70519 | Vegetables: lettuce (lactuca sativa), (other than cabbage lettuce), fresh or chilled | 1 | 0 | 0 | 0 |
| 70610 | Vegetables, root: carrots and turnips, fresh or chilled | 1 | 0 | 0 | 0 |
| 70690 | Vegetables, root: salad beetroot, salsify, celeric, radishes and similar edible roots, fresh or chilled | 1 | 0 | 0 | 0 |
| 70700 | Vegetables: cucumbers and gherkins, fresh or chilled | 1 | 0 | 0 | 0 |
| 70810 | Vegetables, leguminous: peas (pisum sativum), shelled or unshelled, fresh or chilled | 1 | 0 | 0 | 0 |
| 70820 | Vegetables, leguminous: beans (vigna spp., phaseolus spp.), shelled or unshelled, fresh or chilled | 1 | 0 | 0 | 0 |
| 70890 | Vegetables, leguminous: (other than peas and beans), shelled or unshelled, fresh or chilled | 1 | 0 | 0 | 0 |
| 70910 | Vegetables: globe artichokes, fresh or chilled | 1 | 0 | 0 | 0 |

**Table A4.** *Cont.*

| HS6 | HS6 Name | Vegetables (Fresh) | Vegetables (Preserved/ Frozen/Dried) | Legumes (Dried/ Preserved) | Fruit (Fresh and Preserved/ Frozen/Dried) |
|---|---|---|---|---|---|
| 70920 | Vegetables: asparagus, fresh or chilled | 1 | 0 | 0 | 0 |
| 70930 | Vegetables: aubergines, (egg plants), fresh or chilled | 1 | 0 | 0 | 0 |
| 70940 | Vegetables: celery (other than celeriac), fresh or chilled | 1 | 0 | 0 | 0 |
| 70951 | Vegetables: mushrooms, fresh or chilled | 1 | 0 | 0 | 0 |
| 70960 | Vegetables: fruits of the genus capsicum or of the genus pimenta | 1 | 0 | 0 | 0 |
| 70970 | Vegetables: spinach, New Zealand spinach and orache spinach (garden spinach), fresh or chilled | 1 | 0 | 0 | 0 |
| 71030 | Vegetables: spinach, New Zealand spinach and orache spinach (garden spinach), uncooked or cooked by steaming or boiling in water, frozen | 0 | 1 | 0 | 0 |
| 71090 | Vegetable mixtures: uncooked or cooked by steaming or boiling in water, frozen | 0 | 1 | 0 | 0 |
| 71220 | Vegetables: onions, whole, cut, sliced, broken or in powder but not further prepared, dried | 0 | 1 | 0 | 0 |
| 71230 | Vegetables: mushrooms and truffles, whole, cut, sliced, broken or in powder but not further prepared, dried | 0 | 1 | 0 | 0 |
| 200110 | Vegetable preparations: cucumbers and gherkins, prepared or preserved by vinegar or acetic acid | 0 | 1 | 0 | 0 |
| 200190 | Vegetable preparations: vegetables, fruit, nuts and other edible parts of plants, prepared or preserved by vinegar or acetic acid (excluding cucumbers, gherkins and onions) | 0 | 1 | 0 | 0 |
| 200210 | Vegetable preparations: tomatoes, whole or in pieces, prepared or preserved otherwise than by vinegar or acetic acid | 0 | 1 | 0 | 0 |
| 200290 | Vegetable preparations: tomatoes, (other than whole or in pieces), prepared or preserved otherwise than by vinegar or acetic acid | 0 | 1 | 0 | 0 |
| 200310 | Vegetable preparations: mushrooms, prepared or preserved otherwise than by vinegar or acetic acid | 0 | 1 | 0 | 0 |
| 200490 | Vegetable preparations: vegetables and mixtures of vegetables (excluding potatoes), prepared or preserved otherwise than by vinegar or acetic acid, frozen | 0 | 1 | 0 | 0 |
| 200510 | Vegetable preparations: homogenised vegetables, prepared or preserved otherwise than by vinegar or acetic acid, not frozen | 0 | 1 | 0 | 0 |

**Table A4.** *Cont.*

| HS6 | HS6 Name | Vegetables (Fresh) | Vegetables (Preserved/ Frozen/Dried) | Legumes (Dried/ Preserved) | Fruit (Fresh and Preserved/ Frozen/Dried) |
|---|---|---|---|---|---|
| 200560 | Vegetable preparations: asparagus, prepared or preserved otherwise than by vinegar or acetic acid, not frozen | 0 | 1 | 0 | 0 |
| 200570 | Vegetable preparations: olives, prepared or preserved otherwise than by vinegar or acetic acid, not frozen | 0 | 1 | 0 | 0 |
| 71021 | Vegetables, leguminous: peas (pisum sativum), shelled or unshelled, uncooked or cooked by steaming or boiling in water, frozen | 0 | 0 | 1 | 0 |
| 71022 | Vegetables, leguminous: beans (vigna spp., phaseolus spp.), shelled or unshelled, uncooked or cooked by steaming or boiling in water, frozen | 0 | 0 | 1 | 0 |
| 71029 | Vegetables, leguminous: (other than peas or beans), shelled or unshelled, uncooked or cooked by steaming or boiling in water, frozen | 0 | 0 | 1 | 0 |
| 71310 | Vegetables, leguminous: peas (pisum sativum), shelled, whether or not skinned or split, dried | 0 | 0 | 1 | 0 |
| 71320 | Vegetables, leguminous: chickpeas (garbanzos), shelled, whether or not skinned or split, dried | 0 | 0 | 1 | 0 |
| 71331 | Vegetables, leguminous: beans of the species vigna mungo (l.) hepper or vigna radiata (l.) wilczek, dried, shelled, whether or not skinned or split | 0 | 0 | 1 | 0 |
| 71332 | Vegetables, leguminous: small red (adzuki) beans (phaseolus or vigna angularis), shelled, dried, whether or not skinned or split | 0 | 0 | 1 | 0 |
| 71333 | Vegetables, leguminous: kidney beans, including white pea beans (phaseolus vulgaris), dried, shelled, whether or not skinned or split | 0 | 0 | 1 | 0 |
| 71340 | Vegetables, leguminous: lentils, shelled, whether or not skinned or split, dried | 0 | 0 | 1 | 0 |
| 71350 | Vegetables, leguminous: broad beans (vicia faba var. major) and horse beans (vicia faba var. equina and vicia faba var. minor), dried, shelled, whether or not skinned or split | 0 | 0 | 1 | 0 |
| 200540 | Vegetable preparations: peas (pisum sativum), prepared or preserved otherwise than by vinegar or acetic acid, not frozen | 0 | 0 | 1 | 0 |
| 200551 | Vegetable preparations: beans, shelled, prepared or preserved otherwise than by vinegar or acetic acid, not frozen | 0 | 0 | 1 | 0 |
| 200559 | Vegetable preparations: beans, (not shelled), prepared or preserved otherwise than by vinegar or acetic acid, not frozen | 0 | 0 | 1 | 0 |

**Table A4.** *Cont.*

| HS6 | HS6 Name | Vegetables (Fresh) | Vegetables (Preserved/ Frozen/Dried) | Legumes (Dried/ Preserved) | Fruit (Fresh and Preserved/ Frozen/Dried) |
|---|---|---|---|---|---|
| 80410 | Fruit, edible: dates, fresh or dried | 0 | 0 | 0 | 1 |
| 80430 | Fruit, edible: pineapples, fresh or dried | 0 | 0 | 0 | 1 |
| 80440 | Fruit, edible: avocados, fresh or dried | 0 | 0 | 0 | 1 |
| 80450 | Fruit, edible: guavas, mangoes and mangosteens, fresh or dried | 0 | 0 | 0 | 1 |
| 80510 | Fruit, edible: oranges, fresh or dried | 0 | 0 | 0 | 1 |
| 80520 | Fruit, edible: mandarins (including tangerines and satsumas), clementines, wilkings and similar citrus hybrids, fresh or dried | 0 | 0 | 0 | 1 |
| 80530 | Fruit, edible: lemons (citrus limon, citrus limonum), limes (citrus aurantifolia) | 0 | 0 | 0 | 1 |
| 80540 | Fruit, edible: grapefruit, fresh or dried | 0 | 0 | 0 | 1 |
| 80590 | Fruit, edible: citrus fruit not elsewhere specified, fresh or dried | 0 | 0 | 0 | 1 |
| 80610 | Fruit, edible: grapes, fresh | 0 | 0 | 0 | 1 |
| 80620 | Fruit, edible: grapes, dried | 0 | 0 | 0 | 1 |
| 80710 | Fruit, edible: melons (including watermelons), fresh | 0 | 0 | 0 | 1 |
| 80810 | Fruit, edible: apples, fresh | 0 | 0 | 0 | 1 |
| 80820 | Fruit, edible: pears and quinces, fresh | 0 | 0 | 0 | 1 |
| 80910 | Fruit, edible: apricots, fresh | 0 | 0 | 0 | 1 |
| 80920 | Fruit, edible: cherries, fresh | 0 | 0 | 0 | 1 |
| 80930 | Fruit, edible: peaches including nectarines, fresh | 0 | 0 | 0 | 1 |
| 80940 | Fruit, edible: plums and sloes, fresh | 0 | 0 | 0 | 1 |
| 81010 | Fruit, edible: strawberries, fresh | 0 | 0 | 0 | 1 |
| 81020 | Fruit, edible: raspberries, blackberries, mulberries and loganberries, fresh | 0 | 0 | 0 | 1 |
| 81090 | Fruit, edible: fruits not elsewhere specified, fresh | 0 | 0 | 0 | 1 |
| 81110 | Fruit, edible: strawberries, uncooked or cooked by steaming or boiling in water, frozen, whether or not containing added sugar or other sweetening matter | 0 | 0 | 0 | 1 |
| 81120 | Fruit, edible: raspberries, blackberries, mulberries, loganberries, black, white or red currants and gooseberries, uncooked or cooked, whether or not containing added sugar or other sweetening matter | 0 | 0 | 0 | 1 |
| 81310 | Fruit, edible: apricots, dried | 0 | 0 | 0 | 1 |
| 81320 | Fruit, edible: prunes, dried | 0 | 0 | 0 | 1 |
| 81330 | Fruit, edible: apples, dried | 0 | 0 | 0 | 1 |

**Table A5.** Solomon Islands policy documents included in the analysis.

| Food System Sector | Policy Name |
|---|---|
| Agriculture and livestock | Solomon Islands Agriculture Sector Growth and Investment Plan (2021–2030)<br>Agriculture Extension Policy (2017–2021)<br>Solomon Islands Indigenous Fruit and Nut Industry Policies and Strategies (2014–2020) |
| Commerce and industries | Micro, Small and Medium Enterprises (SMEs) Policy and Strategy (2012)<br>Ministry of Commerce, Industry, Labour and Immigration Corporate Plan (2020–2024) |
| Finance, trade and investment | Trade Policy Framework (2015)<br>Solomon Islands Trade Policy Statement (2015) |
| Infrastructure and planning | National Infrastructure Development Plan (2013–2023) |
| Health and education | Lokol Kaikai Initiative (2019–2023) |
| Overarching | Solomon Islands National Development Strategy (2016–2035) |

**Table A6.** Total value of annual household fruit and vegetable acquisition (SBD/year) in urban and rural areas by acquisition (source: 2013 HIES).

| | Food Item | Average Total Value Acquired Per Annum (SBD) | Cash | Home Produced | Gifts Received |
|---|---|---|---|---|---|
| Urban | Fruit | 22,992,496 | 66% | 24% | 9% |
| | Non-starchy vegetables | 49,996,336 | 80% | 16% | 4% |
| Rural | Fruit | 74,750,594 | 10% | 82% | 8% |
| | Non-starchy vegetables | 167,228,659 | 12% | 84% | 5% |

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
