# Peer review of "Fruit and Non-Starchy Vegetable Acquisition and Supply in Solomon Islands: Identifying Opportunities for Improved Food System Outcomes"

_sustainability, doi:10.3390/su15021742_

Round 1

Reviewer 1 Report

The article extensively describes Solomon Island's fruit and non-starchy vegetable acquisition based on data and policy system changes. Overall, the article is good. I would prefer a denser presentation with a shorter length, strengthening the article's brevity. 

Reviewer 2 Report

Main Comments

1.       This study used data from different sources, but there should be a brief description of the sampling framework for the data set. Are the data set lumped/aggregated or fruit/vegetable specific? These and more additional information about the composition of the data set are required.

2.       The analytical techniques and estimation strategies are too vague. The author did not explain how all the adopted data from the different data sources were analyzed. What type of analytical techniques did the authors adopt? It is good to show explicitly the empirical model, and how the included variables are related. For instance, in the multi-logistic analysis, what are the dependent and independent variables?

3.       Page 5, lines 186-187…..The empirical models of all these stated analytical models should be provided--- bi-variatee analysis models with each input variable and the outcome variable; ii. multivariateanalysis’

4.       All the variables used in the model should be explicitly mentioned and defined in Tabular form.

5.       The discussion on page 4, sub-Section 2.2, lines 160 to 174 should be presented in tabular form for clarity.

6.       Were all these data from different sources analyzed separately? How did the author deal with data from different locations?.

7.       The author needs to show the descriptive statistics of all the variables included in the analyses.

Minor comments

1.        There are some typographical errors that need to be checked e.g page 3 lines 131-132 ‘, and  there is need for need for analysis’

2.       Table 2 should be labeled ‘data type and sources’

3.       I found Table 4 on page 10 very confusing and difficult to understand. Maybe it can be better presented, especially the last column titled policy Analysis and framing. Can it be presented in a landscape, instead of a portrait?

Reviewer 3 Report

The paper entitled “Fruit and non-starchy vegetable acquisition and supply in the Solomon Islands: identifying opportunities for improved food system outcomes” showed a whole-country picture of the role and opportunities offered by fruit and non-starchy vegetables in the Solomon Islands. The research aims to identify possible pathways for increasing FNSV consumption to improve human health outcomes – while considering environmental sustainability and identifying opportunities to meet corresponding SDGs. It is found that domestic production of fruit and non-starchy vegetables is insufficient to meet per capita requirements and per capita, national-level supply through imports is inconsequential, highlighting important undersupply issues for the nation. The paper is playing a role in suggesting a policy against the food supply problem. It requires the following revisions before publication.

  1. The appropriate reference numbers can be given in the Data source column of table 1
  2. A nomenclature section can be added for ease of the readers. 
  3. What is the need (motivation) of the presented work for the world in different scenarios and why the study area is selected? 
  4. The scope and limitations of the presented work can be further highlighted. 
  5. Isn’t it good to add a picture or graph to represent the study area?
  6. 2.3 Food Chain: It is suggested to add some pictures and graphs of the vendor survey to support your text.
  7. Improve the quality of Table 3 and add references in the title as well.
  8. For comparison in HIES, FAOSTAT, and PFTD, it would be better to keep all graphs in the same format i.e., line graph or bar graph (if possible)
  9. Make the headings and subheadings of the document clearer. For example, heading 2.2 can have subheadings as 2.2.1 & 2.2.2 for both headings respectively.
  10. Tables 4 and 5 are very important to support the study. Must organize them properly or try another format to represent the data.
  11. Try to improve the language of the whole document from start to finish to give a final finish.
  12. The data in the tables is very unorganized and not easily understandable. Table 3 is a figure, please use the table format of the data 

The research is conducted on a very good topic and has the potential to make a productive policy for the regions having the same problem regarding food supply. I wish the authors the best of luck with their publication. 

Round 2

Reviewer 2 Report

No additional comments.